# Analysis of Suicidal Behavior in Adult Inpatients with Anorexia Nervosa: Prevalence of Suicide Attempts and Non-Suicidal Self-Injury and Associated Factors—Data Before and After the COVID-19 Pandemic

**DOI:** 10.3390/jcm13226952

**Published:** 2024-11-18

**Authors:** Paola Longo, Matteo Martini, Federica Toppino, Carlotta De Bacco, Antonio Preti, Giovanni Abbate-Daga, Matteo Panero

**Affiliations:** Eating Disorder Center for Treatment and Research, Department of Neuroscience, University of Turin, Via Cherasco 11, 10126 Turin, Italy; paola.longo@unito.it (P.L.); matteo.martini@unito.it (M.M.); federica.toppino@unito.it (F.T.); carlideb@yahoo.it (C.D.B.); antonio.preti@unito.it (A.P.); matteo.panero@unito.it (M.P.)

**Keywords:** suicidality, eating disorders, suicide, self-harm, psychopathology, COVID-19

## Abstract

**Background/Objectives**: Anorexia nervosa (AN) has a high mortality rate frequently related to suicidality; however, there are few studies on suicide attempts (SAs) and non-suicidal self-injuries (NSSIs) in adult inpatients with AN. This study aims to describe SA and NSSI prevalence and related clinical and sociodemographic factors in adult inpatients with AN. **Methods**: We retrospectively analyzed data on 298 inpatients hospitalized between 2014 and 2023. Suicidality and clinical and sociodemographic data were collected by experienced psychiatrists; then, the patients completed a battery of self-report questionnaires investigating eating-related and general psychopathology. **Results**: A total of 9.7% of the inpatients reported an SA in their lifetime, and 13.4% reported NSSI. The percentages were lower among patients with restricter-type AN (5.6% SA and 6.6% NSSI) and higher among patients with binge–purging AN (18% SA and 27% NSSI). SAs were associated with unemployment, binge–purging AN, personality disorders, and lifetime sexual abuse; NSSIs were correlated with family psychiatric disorders, binge–purging AN, personality disorders, body dissatisfaction, restriction, and eating-related concerns. No differences emerged in the frequency of reported suicidality between patients tested before and after the COVID-19 pandemic. **Conclusions**: Suicidality is a relevant issue in AN. Many factors in the history of the patients and their eating-related pathologies should be considered as potentially associated with SA and NSSI and carefully assessed.

## 1. Introduction

Anorexia nervosa (AN) is characterized by abnormal eating behaviors leading to a critically low weight and important medical and psychosocial consequences [1]. Phenotypically, AN has two diagnostic subtypes: restricter AN (R-AN), in which patients lose weight through diets and physical activity, and binge–purging AN (BP-AN), characterized by binge episodes followed by compensatory conducts such as vomiting or laxatives use. Clinicians often describe patients with AN as resistant and difficult to treat, leading to feelings of inadequacy and vulnerability [2], and making AN a heavy burden for families, patients, and the National Health System. The burden is worsened by a high rate of mortality [3,4]; death in AN is not infrequently related to suicidal behaviors [5]. Relatedly, beyond the diagnostic criteria, patients with AN assign subjective meanings to anorectic behaviors that extend beyond the socioculturally shared, weight-centered paradigm. These meanings can encompass several conceptions of the illness, including association with death (i.e., wishing to starve oneself to death), to the extent that patients conceptualize AN as a way to disappear [6,7]. Moreover, patients with AN experience a wide range of negative emotions potentially connected to suicidality: for instance, the commonly referred feelings of shame can contribute to the desire not only to change body shape, but also to disappear in terms of suicide [8]. The relationship between suicidality and eating disorders is well established. Many studies have tried to describe suicidality in eating disorders and in AN: meta-analyses reported that suicide attempts (SAs) are more frequent in patients with AN than in the general population, and overall, lethal SAs are high in AN [9,10,11]. Data on the prevalence of SAs are heterogeneous: a retrospective study found that 6.7% of patients with AN and bulimia nervosa attempted suicide, requiring hospitalization [12], while a recent meta-analysis described a prevalence of SAs of 22% [11]. Relatedly, Soullane and colleagues (2023) showed that young patients hospitalized for SA had a 5.5 times higher risk of hospitalization for an eating disorder in the following year compared with those without SA history [13]. These studies, however, are not specific for AN. A meta-analysis by Mandelli and colleagues (2019) found a proportion of SAs in AN of 12.5% [14]. An important variable in determining the frequency of SAs is the diagnostic subtype of AN: the literature has confirmed a higher proportion of SAs in BP-AN than in R-AN [14,15], and purging behavior was estimated as a significant predictor of the presence or absence of suicidal ideation [16]. Furthermore, non-suicidal self-injury (NSSI) is commonly reported by patients with eating disorders, especially if affected by bulimia nervosa or BP-AN [14], as recently confirmed by Kirkpatrick and colleagues [17] and Arnold et al. [18]. A recent meta-analysis confirmed the higher risk of suicidality in people with eating disorders compared with healthy participants but not compared with other psychiatric disorders [19]; interestingly, the authors did not find differences among the specific eating disorder diagnoses, in contrast with the above-cited studies. These data add heterogeneity to the literature on suicidality in eating disorders and in AN specifically, highlighting the need for further investigations.

The majority of the recent studies on suicidality and NSSI in AN were conducted on samples of young or adolescent patients: in very young people, the rate of SAs ranges from 6% to 37.2% [15,20]; concerning NSSI, variability is even higher, with a prevalence ranging from 14% to 68% [21,22]. The method used to measure NSSI likely contributes to the observed variance.

The research on this topic has focused on clarifying the potential risk and associated factors of suicidality in AN. The variables explored in the literature can be clustered into three classes: individual/sociodemographic factors, clinical AN-related aspects, and personality traits or comorbid psychiatric symptoms. Studies focusing on the first class of factors found that childhood abuse, especially physical and sexual abuse, are important risk factors for SA in adulthood [5]; the same study showed a role of familiarity in eating disorders [5].

Concerning clinical AN-related factors, some researchers identified specific eating-related aspects associated with SA risk such as longer illness duration, lower body mass index (BMI), a higher number of past treatments, bulimic and/or purging symptoms, and a diagnostic switch from restricting to binge–purging AN [14,23,24,25]. Furthermore, body dissatisfaction was described as associated with SA [24]: this should be kept particularly in mind, as body dissatisfaction is one of the core symptoms of AN. Finally, there are personality-related and comorbid pathological factors associated with suicidality in AN such as substance and alcohol abuse, depression, impulsivity, and emotion dysregulation [5,23,25].

As regards NSSI, it was estimated that AN patients with NSSI showed higher eating-related symptom severity and a longer illness history; moreover, these patients could present traits typical of borderline personality disorder, frequent psychiatric comorbidity, and the use of psychiatric medications [26,27,28]. Finally, shame is also connected to NSSI as a form of self-humiliation [8].

It must be specified that recent studies have focused mainly on adolescent samples, often diagnostically heterogeneous; thus, there are scarce data on adults with AN. Furthermore, there is evidence of an increase in SAs [29] and in NSSIs [30] in adolescents with EDs after the COVID-19 pandemic; however, these data are still to be clarified in EDs in the adult population. Relatedly, the studies on suicidality in Italy during and after the pandemic reflected a non-linear trend: there is accordance on the decrease in SAs and suicide ideation during the lockdown compared with the data collected before the COVID-19 pandemic [31,32], while fewer data are available on the post-pandemic suicidality rate showing a slight increase in SA and suicide ideation and planning [31,32,33]. Further data on COVID-19 are needed, especially considering the research finding higher levels of depression and anxiety during the pandemic in individuals with pre-existing co-occurrence of self-harm and disordered eating compared with those without these symptoms [34].

In light of these scarce data from the adult population, and the likely impact of the COVID-19 pandemic, we aimed to carry out a retrospective study on suicidality in adult inpatients with AN referred to our center from 2014 to 2023. We set the following objectives: (a) to estimate the lifetime prevalence of SAs and NSSIs in the whole sample and in the AN subtypes; (b) to describe individual/sociodemographic factors, clinical AN-related aspects, and personality traits or comorbid psychiatric symptoms of patients with and without SAs or NSSIs in order to provide a picture of patients with AN and a history of suicidality; (c) to explore the association between SA or NSSI and several variables (considered based on the previous literature) clustered in five classes, namely, sociodemographic variables, clinical eating-related variables, comorbid symptoms and psychiatric disorders, eating-related psychopathology, and general psychopathology; (d) to describe the prevalence of SAs and NSSIs before and after the COVID-19 pandemic.

According to the previous studies, we expected a higher prevalence of SA and NSSI in BP-AN than in R-AN and a stronger association of suicidality with illness duration, bulimic symptoms, family eating disorders, and depression. Based on the data in the adolescent population, we expected an increase in SAs and NSSIs after the COVID-19 pandemic.

## 2. Materials and Methods

### 2.1. Participants

The study sample includes 298 inpatients with AN who attended the Eating Disorder Center of the “Città della Salute e della Scienza” hospital of the University of Turin, Italy, between 2014 and 2023. Our center offers different levels of care based on the severity of the illness phase: patients in a very acute phase of the illness are hospitalized. The recruitment used the following inclusion criteria: (a) formal diagnosis of AN according to the Structured and Clinical Interview for DSM-5 (SCID-5; [35]) and (b) age between 18 and 65 years. Exclusion criteria were as follows: (a) comorbid psychotic disorders, (b) neurological diseases (e.g., epilepsy, traumatic brain injury), and (c) cognitive impairment. During the first ambulatory visit or the first day of hospitalization, patients were instructed about the study recruitment. All participants agreed to fill the proposed questionnaires.

All patients signed an informed consent; the study was approved by the ethical committee of the University of Turin on 19 November 2018 under the registration number CS2/1004.

### 2.2. Procedure and Materials

An experienced psychiatrist interviewed all patients during the first day of hospitalization to confirm the diagnosis of AN and to collect sociodemographic data (including lifetime sexual abuse) and personal history of suicidal attempts (SAs) and non-suicidal self-injuries (NSSIs). For patients admitted more than once between 2014 and 2023, data on the first hospital admission were considered for this study. Patients were asked to complete the following self-report questionnaires during the first week of hospitalization:

The Eating Disorders Inventory-2 (EDI-2; [36]) is one of the most-used self-report tools to assess eating-related core and associated psychopathology. EDI-2 is also widely used in studies on suicidality in eating disorders [37]. Sixty-four items measure cognitive and behavioral features typical of bulimia and AN. The scoring is organized into 11 subscales (drive for thinness, bulimia, body dissatisfaction, perfectionism, asceticism, fear of maturity, ineffectiveness, interpersonal distrust, impulsivity, interoceptive awareness and social insecurity). In the present study, we considered the first three subscales, reflecting the core eating-related symptoms of AN. The questionnaire has a good internal consistency (Cronbach alpha values ranging from 0.83 to 0.93 [36]).

The Eating Disorder Examination Questionnaire (EDE-Q; [38]) is among the most-used tools assessing disordered eating behavior and comorbidity, including suicidality [39,40]. Participants are asked to answer 28 items investigating eating-related symptoms and behavior in the last 28 days. A global score and four subscales are provided: eating restraint, food concerns, weight concerns, and shape concerns. The internal consistency of the Italian version is good, with Cronbach alpha values > 0.90 [41].

We administered the State-Trait Anxiety Inventory (STAI; [42]) tool to assess anxiety levels: 20 questions investigate trait anxiety (i.e., stable levels of anxiety), while the other 20 reflect state anxiety (i.e., current levels of anxiety). Cronbach alpha values ranging from 0.86 to 0.95 indicate good internal consistency [43]. The questionnaire has the advantage of giving an idea of both the current and the usual anxiety, and it is also used to investigate anxiety in relation to suicidality [44].

The Beck Depression Inventory (BDI; [45]) is one of the most-validated questionnaires for depression, frequently adopted to study the association between depression and suicidality [37,46]. Depressive symptoms are assessed with 21 questions. The global score describes different levels of depression: low (scores from 0 to 4), moderate (scores from 5 to 15), and severe (scores from 16 to 39). The internal consistency is good, with a Cronbach alpha value of 0.86 [47].

### 2.3. Statistical Analysis

We used the SPSS 28.0 statistical software package (IBM SPSS Statistics for Windows, Version 28.0 Armonk, NY, USA: IBM Corp). Descriptive analyses (i.e., frequencies and means with standard deviation) were adopted to describe the sample. A Mann–Whitney test was conducted to investigate the differences in continuous variables between patients with and without SA and with and without NSSI; differences between groups in categorical variables were investigated with the chi-squared test. Then, for both SA and NSSI, we conducted 5 regression models to assess the association between SA or NSSI and sociodemographic variables, clinical eating-related variables, comorbid symptoms and psychiatric disorders, eating-related psychopathology, and general psychopathology. In particular, binary logistic regression was used to test the association between sociodemographic or clinical variables and the outcome: the outcome was binary, depending on the presence (yes = 1) or absence (no = 0) of lifetime SA or NSSI. The predictors were entered as continuous or categorical according to the nature of the variable (e.g., BMI = continuous; presence or absence of psychiatric comorbidity = categorical). The regression results were exposed, reporting the odd ratios with a 95% confidence interval.

We then analyzed how the two phenomena developed across the years, graphically representing the percentage of patients reporting SA and NSSI per year, and we used the Fisher exact test to investigate the difference in the distribution of SA and NSSI before and after the COVID-19 pandemic. Data collection considered how many patients in a year reported a lifetime SA or lifetime occurrence of NSSI.

## 3. Results

The sample included 298 patients with AN; 66.4% of the sample (N = 198) was affected by R-AN, while 33.6% (N = 100) by BP-AN. The majority of the participants were female (94.6%; N = 282), and men represented 5.4% of the sample (N = 16).

SA was reported by 29 patients (9.7%) and NSSI by 40 subjects (13.4%).

Considering patients with R-AN, the prevalence of SA was 5.6%, and that of NSSI was 6.6%. Among patients with BP-AN, 18% reported a lifetime SA and 27% NSSI, with chi-squared showing a significant difference between the two diagnostic groups in the frequency of both SA (*p* < 0.001) and NSSI (*p* < 0.001).

### 3.1. Sociodemographic and Clinical Characteristics of Patients with and Without SA

Participants reporting lifetime SA were mostly BP-AN, and had a higher BMI and lowest lifetime weight, compared with those without SA history. A higher frequency of NSSI, sexual abuse, and personality disorders appeared in the SA group (Table 1). Data on the differences between patients with and without SA on psychometric scales are available in the Appendix A: patients with SA scored higher than those without lifetime SA in bulimia and ineffectiveness subscales of EDI-2, weight concern (EDE-Q), and anxiety and depression scores measured with STAI and BDI, respectively.

### 3.2. SA Associated Variables

Among the explored sociodemographic variables, just occupation was statistically associated with lifetime SA, namely, unemployed patients had a higher probability of reporting SA (Model 1; Table 2). As regards clinical eating-related features, patients with BP-AN had an almost threefold higher likelihood of reporting lifetime SA compared with those with R-AN (Model 2; Table 2). Comorbid personality disorders and lifetime sexual abuse were positively and significantly associated with SA (Model 3; Table 2). As regards eating-related and general psychopathology, we included in the model those variables significantly different between patients with and without SA: none of the considered variables were significantly associated with SA (Models 4 and 5; Table 2).

### 3.3. Sociodemographic and Clinical Characteristics of Patients with and Without NSSI

Patients reporting NSSI were younger at illness onset than patients without NSSI. Among patients with NSSI emerged a significantly higher percentage of subjects with BP-AN, with higher BMI, under psychopharmacological treatment, with lifetime SA, with a history of sexual abuse, and with personality disorders than among those without NSSI; moreover, patients with NSSI showed more frequently a family history of psychiatric disorders (Table 3). Data on the differences between patients with and without NSSI on psychometric scales are available in the Appendix A: patients with lifetime NSSI scored higher than those without NSSI in all the EDI-2 subscales but perfectionism and maturity fear, in all the EDE-Q subscales, and in the anxiety and depression scores obtained with STAI and BDI, respectively.

### 3.4. NSSI-Associated Variables

Family history of psychiatric disorders was the only sociodemographic variable associated with NSSI (Model 1b; Table 4). Patients with BP-AN and higher BMI were more likely to report NSSI (Model 2b; Table 4). Comorbid personality disorders were also significantly associated with NSSI (Model 3b; Table 4). Regarding eating-related and general psychopathology, we included in the model those variables that significantly differed between patients with and without NSSI: positive significant associations were found between NSSI and the body dissatisfaction subscale of EDI-2 and the restraint, food concern, and weight concern subscales of EDE-Q (Model 4b; Table 4).

### 3.5. Temporal Trend of SA and NSSI

No differences emerged between patients tested before and after the COVID-19 pandemic in SA (*p* = 0.366; frequency of SA before COVID-19 = 10.8%; frequency of SA after COVID-19 = 7.4%) and NSSI (*p* = 0.554; frequency of NSSI before COVID 19 = 14.2%; frequency of NSSI after COVID 19 = 11.7%). Figure 1 reports the temporal distribution of SA and NSSI across 2014–2023.

## 4. Discussion

The present study aimed to provide recent data on suicidality in adult subjects with AN. The following main findings emerged: (1) In the whole sample of adult patients with AN, 9.7% reported a lifetime SA and 13.4% reported NSSI; the percentages were lower when considering inpatients with R-AN (5.6% SA and 6.6% NSSI) and higher among inpatients with BP-AN (18% SA and 27% NSSI). (2) The regression models showed a significant association between SA and occupation, namely, the morelonger patients were unemployed, the more likely they were to report an SA. The models also showed an association with AN subtype (more SAs were reported by patients with BP-AN than with R-AN), and positive associations also emerged between SA and comorbid personality disorders and sexual abuse. (3) NSSI was significantly associated with BP-AN, BMI, family psychiatric disorders, comorbid personality disorders, body dissatisfaction, restriction, and weight and food concern. (4) Neither the frequency of SAs nor that of NSSIs underwent a significant trend change with the COVID-19 pandemic.

The overall prevalence of SA and NSSI in this sample was lower compared with many previous studies [15,20] that focused on adolescent patients; however, our data on the higher presence of SA and NSSI in the bulimic variants of AN are in accordance with Arnold et al. (2023) and Kirkpatrick et al. (2023). These two studies, however, focused on a sample of young patients and samples of individuals with very heterogeneous mean age (i.e., the review collected studies with mean age ranging from 13.8 to 33.4), respectively, and described percentages of SA and NSSI higher than ours [17,28]. Differently, our numbers are similar to those found by Mandelli and colleagues and Cliffe et al., who recruited adult patients [12,14]. Moreover, it should be considered that the majority of our patients were diagnosed with R-AN; thus, analyzing separately the two AN subtypes, patients with BP-AN reported a much higher frequency of SA and NSSI. The role of the binge–purging subtype in determining a greater predisposition to SA in our study is strengthened by the significant association between SA and BP-AN. These data are in line with the previous literature describing a higher prevalence of suicidality in bulimic variants of eating disorders and binge–purging symptoms as significant predictors of SA [16]. Differently, this study is in contrast with the meta-analysis by Sohn and colleagues [19]. Relatedly, our study found a positive association between NSSI and BP-AN and bulimic symptoms, in line with previous studies. To explain this association, some authors proposed that repeated engagement in painful binge–purging behaviors could predispose patients to severe NSSI through habituation to pain and fear [14]. Interestingly, pain tolerance is one of the suicide risk factors described by Joiner in the interpersonal model [48,49]; this mechanism could mediate an indirect relation between SA and NSSI. Future longitudinal studies are needed to confirm this hypothesis.

A factor significantly associated with both SA and NSSI is a comorbid personality disorder. The literature has described an association between suicidality and psychiatric comorbidity [28,50]. The most frequent personality disorder diagnosis in our sample was borderline personality disorder. This is in line with the literature showing high rates of suicide and self-harm in patients with both AN and borderline personality disorders [51,52]. Moreover, impulsivity and affective dysregulation appeared to be linked to suicidality, especially to NSSI [53,54]: these features are particularly pronounced in borderline personality disorder. Despite the cited studies seeming to suggest a role of borderline personality in enhancing suicidality in AN, this causal correlation cannot be supported by our data; future longitudinal studies could clarify whether borderline personality disorder worsens suicidality in AN or vice versa.

SA was positively associated with sexual abuse: the association between sexual trauma and SA is well established, and childhood sexual abuse is described as a predictor of both SA and NSSI [55,56]. Our data are in line with studies reporting a link between suicidality and abuse [15,28]. Given the high rates of traumatic events in AN, especially related to sexual abuse [57], and their association with suicidality, an assessment of patients’ traumatic history should always be performed. Interestingly, sexual abuse is more frequent in the binge–purging subtype of AN [57], the phenotype more often associated with suicidality. The link between BP-AN and SA could be partially mediated by the effects of having suffered sexual abuse; however, so far no empirical data have supported these hypotheses, which should be considered cautiously. Longitudinal studies are needed to further explore this speculation. Relatedly, in the case of both borderline personality disorders and childhood trauma, the picture is extremely complex and entangled, and the higher the clinical complexity, the harder the causal relationships to be drawn.

Our results showed a negative association between SA and occupation, namely, patients without a job were more prone to report a lifetime SA: to our knowledge, this is the first study to explore this association in AN. Previous studies on suicide reported, however, a negative association between employment and suicidality in the general population in Europe: SAs were related to the risk of unemployment and low financial expectations [58,59].

Among sociodemographic factors, family psychiatric disorders were negatively and significantly associated with NSSI. This is the first study to find this specific link in AN; this datum is novel, and further studies should deepen this link. However, it could be hypothesized that growing up or living with a family member affected by a psychiatric disorder could lead to increased awareness and resistance toward anti-conservative behaviors.

NSSI was also associated with body dissatisfaction, a core symptom of AN. This is in line with the previous literature describing body dissatisfaction as a shared psychological feature involved in both AN and NSSI [60] and reporting body dissatisfaction being a predictor of NSSI in both clinical and nonclinical samples [61,62,63]. Relatedly, in a systematic review by Hielscher and colleagues, an association of NSSI was found not only with body dissatisfaction but also with alteration of body ownership and interoception [64]: these impairments are well described in patients with AN [65]. According to the authors of the above-described review, the overall altered body representation could undermine the self-protection mechanism, thus predisposing to self-harm [64]. Moreover, a positive and significant association was also observed between NSSI and the restraint subscales of EDE-Q: this datum is novel and suggests that, despite the higher prevalence of NSSI among patients with bulimic symptoms, restriction should be also considered and monitored as a potential suicidality-related factor. Food concern and weight concern were also significantly related to NSSI, demonstrating a strong relationship between self-harm and symptoms typical of eating disorders: this datum is in line with the previous literature describing more severe eating-related psychopathology in patients with eating disorders who practiced self-harm compared with those without this behavior [66,67]. Taken together, the higher scores on these specific AN symptoms scales confirm how NSSI can be an exacerbated way of managing the fear of gaining weight by acting against the body.

Of note, anxiety and depression were higher in both patients with SA and with NSSI, but they were not significantly associated with SA and NSSI. The lack of significance could be due to the fact that the relationship between suicidality and anxiety and depression in AN is so entangled that anxiety and depression do not result as clear predictors of suicidality, suggesting mixed and diversified pathways linking these factors.

Finally, we analyzed the temporal trend of SA and NSSI. Despite the epidemic of SAs and NSSIs in the adolescent population in the post-pandemic era [29,30], we found no differences in the trend of the two phenomena reported by adult inpatients recruited before and after the COVID-19 pandemic. This is an interesting phenomenon that confirms that the greatest negative impact of the COVID-19 pandemic has been especially on very young people. The lack of increase in reported SAs and NSSIs in our sample of adults with AN could be explained by some hypotheses: firstly, some subjects were already ill before the pandemic and may have been less influenced by the lockdown for daily activities; secondly, adult subjects may have adopted more effective coping strategies during the pandemic, given their age; thirdly, the help provided by our center to our patients quite closely during 2020 may have mitigated the negative effect of the pandemic. Rather, in our previous study, we identified in the adult population an increase in compulsory exercise and body dissatisfaction [68]. Certainly, our results could be affected by a data collection bias; however, it should be remembered that our center is the only public hospitalization center specializing in EDs in a vast area where 4.5 million inhabitants live. Other future studies are necessary on other samples and on the longitudinal trend of SAs and NSSIs over time.

The present study has some limitations: firstly, the design is cross-sectional and based on self-reported data; thus, recall bias cannot be excluded. Secondly, we did not administer specific scales to measure levels of suicidality. Third, some crucial aspects, such as alcohol and substance abuse, previous psychotherapeutic sessions, and suicide ideation, were not specifically investigated. Finally, data on COVID-19 were based on the lifetime frequency of SA and NSSI reported by patients hospitalized for the first time; thus, the analysis was not on the incidence, yet they suggest a stable trend in suicidality before and after the COVID-19 pandemic. Despite these limitations, the study provides recent data on SA and NSSI in a large sample of adult inpatients with AN recruited over a long period. Moreover, we considered SA and NSSI separately, adding knowledge on the two aspects of suicidality that are often considered together [53]. Relatedly, we showed that SA and NSSI display both shared and specific associations: the binge–purging subtype of AN and a comorbid personality disorder are associated with both SA and NSSI, confirming their role in determining a higher risk of suicidality in AN. Differently, occupation was associated with SA only, while family psychiatric disorders and body dissatisfaction were associated specifically with NSSI. Surprisingly, we did not find associations between SA and eating symptoms; this lack of significance should not, however, lead to neglect of anorectic symptoms in relation to suicidality. A qualitative study, indeed, suggested that for some patients, AN itself could be a concrete expression of the desire to disappear and put an end to life; contextually, the authors reported that death by starvation was defined by patients as less brutal compared with other suicide methods [7]. In contrast, NSSI is relevantly associated with a wide range of typical symptoms of eating disorders and body-related disturbances.

In light of this, the present study represents an effort to isolate specific factors in the complex and entangled picture of eating disorders; future studies should keep on parsing out single variables involved in the clinical presentation of the disease, trying to adopt a longitudinal design to seek causative instead of correlational results.

These data confirm the great impact of suicidality in AN and suggest carefully assessing and treating the specific factors associated with a greater risk of SA and NSSI. The study has clinical implications too. The adult population affected by AN remains at high risk of suicide that should always be investigated, especially in the presence of the identified associated factors. Psychiatric comorbidity seems to be a crucial point and is often overlooked by clinicians [69]: greater attention should be paid to an early diagnosis of psychiatric comorbidity. Moreover, the study confirms the importance of the relationship with one’s body for NSSIs: greater consideration should be given to subjects who feel shame and guilt toward their body shape [8]. Furthermore, the presence of less-considered factors such as the absence of employment tells us to focus also on the personal and social effects of the disease, which can lead to fatal feelings of loneliness. Finally, since SA was not related to eating symptoms but to other clinical features, the data suggest focusing on the relevance of “collateral” symptoms in AN, not those strictly related to eating psychopathology. In this context, the lack of association between SA and BMI suggests not to let our guard down even in case of weight restoration; as described by Barko and colleagues, indeed, the weight-centered perspective, for which weight restoration is a measure of recovery, could lead to neglecting other important factors contributing to relapse [6].

All these factors should be considered to improve the prevention and treatment of suicidality in AN: for instance, a careful and exhaustive assessment of the risk of suicide should always be a starting point leading to the choice of the right suicide-focused treatment (e.g., cognitive behavioral therapy, dialectical behavioral therapy [70]), especially in light of the well-known resistance of patients with AN to open up about their feelings [2], which can make it more difficult to investigate emotions potentially related to suicidality. Moreover, an updated and complete knowledge of all factors potentially associated with suicidality in AN is warranted for all clinicians dealing with these issues in order to enhance a rapid and correct recognition of suicide risk. These strategies could, in turn, ameliorate the job-related well-being of therapists and clinicians working with patients with AN and suicidality, since the perceived suicide risk of the patient contributes to depression and overwhelmed feelings reported by psychiatrists [71] and could worsen the reported sensation of vulnerability often experienced by therapists in the field of eating disorders [2].

From a theoretical standpoint, the present study may boost future research on the topic: for instance, studies with a longitudinal design are needed, and future works should deepen the relationships among the factors significantly associated with SA and NSSI. Moreover, studies on suicidality through the lifespan in AN are needed: studies indeed have described not only static factors (e.g., gender) linked to suicide but also dynamic ones (e.g., substance use) that change in the different stages of life [72,73]. For instance, the focus on older adults with AN could be insightful. It is acknowledged, indeed, that suicide in individuals older than 60 is a critical issue [74], and it could be useful to deepen understanding of how suicidality develops in older patients with AN given the potential overlap of risk factors, such as depression or additional cognitive impairment, that could exacerbate the pre-existing deficits in cognitive flexibility and decision-making that are typical of AN and associated with suicide risk in older adults [74]. The exploration of patients’ feelings toward death and suicidality may also be explored with qualitative studies. Finally, investigations on possible biological factors potentially involved in the association between eating disorders and suicidality are scarce.

## Figures and Tables

**Figure 1 jcm-13-06952-f001:**
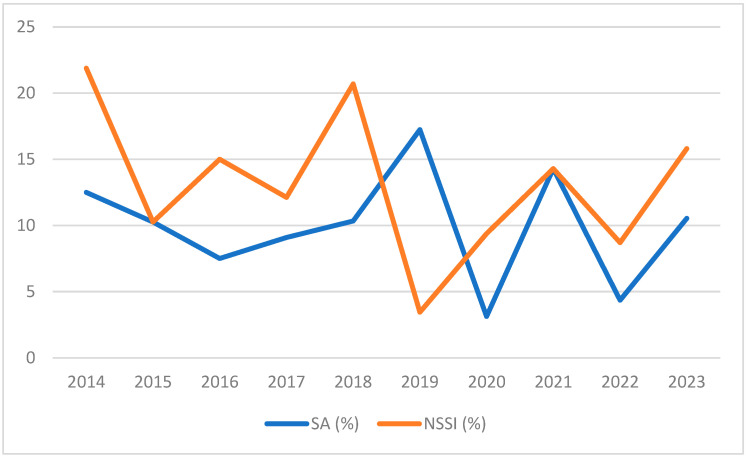
Temporal distribution of SA and NSSI.

**Table 1 jcm-13-06952-t001:** Sociodemographic and clinical characteristics of patients with and without SA.

	Patients with SA (N = 29)	Patients Without SA (N = 269)	Squared chi/Z	*p*	Effect Size Cohen’s d/Phi
Gender	6.9% M 93.1% F	5.2% M 94.8% F	0.148	0.475	−0.022
Age	25.2 (SD 9.7)	25.2 (SD 9.9)	−0.409	0.682	−0.002
Education	13.5 (SD 2.6)	13.5 (SD 2.6)	−0.262	0.793	−0.002
Occupation	58.6% employed or student 41.4% unemployed	75.8% employed or student 24.2% unemployed	4.049	0.071	−0.117
Housing solution	31% independent 69% with family of origin	16.7% independent 81.8% with family of origin	3.922	0.141	0.115
Relationship	58.6% in relationship 41.4% single	23.4% in relationship 76.2% single	4.550	0.103	0.103
AN subtype	37.9% AN-R 62.1% AN-BP	69.5% AN-R 30.5% AN-BP	14.340	< 0.001	0.219
Psychopharmacological treatment	79.3% yes 20.7% no	62.8% yes 37.2% no	3.104	0.078	0.102
Illness duration	6.6 (SD 8.5)	6.8 (SD 8.6)	0.596	0.551	0.020
Age at illness onset	18.4 (SD 7.1)	18.2 (SD 5.5)	−0.435	0.664	−0.038
BMI	15.8 (SD 1.7)	14.4 (SD 2)	−3.744	**<0.001**	−0.701
Caloric intake	691.6 (SD 387.7)	730.8 (SD 398.1)	0.402	0.688	0.098
Lowest lifetime weight	38.4 (SD 5.6)	36.1 (SD 5.9)	−2.156	**0.031**	−0.391
Previous hospitalizations	2.2 (SD 3.5)	1 (SD 2)	−1.792	0.073	−0.561
NSSI	37.9% yes 62.1% no	10.8% yes 89.2% no	16.605	**<0.001**	0.236
Lifetime sexual abuse	27.6% yes 72.4% no	6.3% yes 93.7% no	15.405	**0.001**	0.227
Psychiatric comorbidities	48.3% yes 51.7% no	37.9% yes 62.1% no	1.181	0.187	0.063
Personality disorders	48.3% yes 51.7% no	14.9% yes 85.1% no	19.689	**<0.001**	0.257
Family history of psychiatric disorders	41.4% yes 58.6% no	31.3% yes 68.4% no	1.205	0.186	0.064
Family history of eating disorders	10.3% yes 89.7% no	13.4% yes 86.6% no	0.212	0.455	−0.027

BMI = Body mass index; NSSI = Non-suicidal self-injury.

**Table 2 jcm-13-06952-t002:** Regression models for SA.

	B	SE	OR	95% CI	*p*
Model 1: association between SA and sociodemographic variables					
Education	−0.005	0.074	0.995	0.860–1.152	0.950
Gender	−0.320	0.795	0.726	0.153–3.452	0.688
Occupation	−0.802	0.406	2.230	1.007–4.940	0.048
Family history of psychiatric disorders	−0.485	0.408	0.616	0.277–1.370	0.235
Family history of eating disorders	0.477	0.648	1.612	0.453–5.737	0.461
Model 2: association between SA and eating-related clinical features					
AN subtypes	1.041	0.525	2.833	1.013–7.921	**0.002**
Illness duration	−0.052	0.037	0.949	0.883–1.021	0.160
Age at illness onset	−0.006	0.037	0.994	0.924–1.070	0.872
BMI	0.184	0.161	1.204	0.878–1.650	0.250
Lowest lifetime weight	−0.022	0.051	0.978	0.884–1.082	0.667
Psychopharmacological treatment	1.220	0.665	3.387	0.920–12.468	0.067
Crossing over	0.387	0.738	1.472	0.346–6.258	0.601
Previous hospitalizations	0.190	0.087	1.210	1.019–1.435	0.111
Model 3: association between SA and comorbid clinical variables					
NSSI	0.892	0.510	2.440	0.898–6.630	0.080
Psychiatric comorbidities	0.130	0.440	1.139	0.481–2.701	0.767
Personality disorders	1.333	0.478	3.792	1.486–9.675	**0.007**
Lifetime sexual abuse	1.292	0.568	3.641	1.195–11.091	**0.006**
Model 4: association between SA and eating-related psychopathology					
EDI-2 bulimia	0.021	0.051	1.022	0.925–1.128	0.673
EDI-2 ineffectiveness	0.047	0.039	1.048	0.971–1.132	0.228
EDE-Q weight concerns	0.309	0.205	1.326	0.912–2.036	0.131
Model 5: association between SA and general psychopathology					
BDI	0.018	0.044	1.018	0.934–1.108	0.688
STAI-state	0.015	0.025	1.015	0.966–1.066	0.562
STAI-trait	0.045	0.034	1.046	0.980–1.117	0.177

BMI = Body mass index; NSSI = Non-suicidal self-injury; EDI-2 = Eating Disorder Inventory-2; EDE-Q = Eating Disorder Examination Questionnaire; BDI = Beck Depression Inventory; STAI = State-Trait Anxiety Inventory.

**Table 3 jcm-13-06952-t003:** Sociodemographic and clinical characteristics of patients with and without NSSI.

	Patients with NSSI (N = 40)	Patients Without NSSI (N = 258)	Squared chi/Z	*p*	Effect Size Cohen’s d/Phi
Gender	2.7% M 97.5% F	5.8% M 94.2% F	0.749	0.339	0.050
Age	23.1 (SD 7.9)	25.2 (SD 9.9)	1.093	0.273	0.237
Education	13.2 (SD 2.8)	13.5 (SD 2.6)	0.766	0.443	0.129
Occupation	70% employed or student 30% unemployed	74.8% employed or student 25.2% unemployed	0.417	0.551	−0.037
Housing solution	20% independent 80% with family of origin	17.8% independent 80.6% with family of origin	0.712	0.701	0.049
Relationship	32.5% in relationship 67.5% single	24% in relationship 75.6% single	1.446	0.485	0.070
AN subtype	32.5% AN-R 67.5% AN-BP	71.7% AN-R 28.3% AN-BP	26.884	<0.001	0.300
Psychopharmacological treatment	80% yes 20% no	62% yes 38% no	4.887	**0.033**	0.128
Illness duration	6.9 (SD 8.1)	6.8 (SD 8.6)	−1.334	0.182	−0.023
Age at illness onset	16.4 (SD 7.2)	18.2 (SD 5.5)	2.110	**0.035**	0.371
BMI	15.9 (SD 1.9)	14.4 (SD 2)	−4.700	**<0.001**	−0.853
Caloric intake	640.8 (SD 346.6)	730.8 (SD 398.1)	1.331	0.183	0.253
Lowest lifetime weight	37.6 (SD 5.4)	36.1 (SD 5.9)	−1.663	0.093	−0.256
Previous hospitalizations	1.7 (SD 3.1)	1 (SD 2)	−0.419	0.676	−0.284
Suicide attempts	27.5% yes 72.5% no	7% yes 93% no	16.605	**<0.001**	0.236
Lifetime sexual abuse	20% yes 80% no	6.6% yes 93.4% no	8.104	**0.010**	0.165
Psychiatric comorbidities	52.5% yes 47.5% no	36.8% yes 63.2% no	3.581	0.080	0.110
Personality disorders	57.5% yes 42.5% no	12% yes 88% no	42.288	**<0.001**	0.403
Family history of psychiatric disorders	50% yes 50% no	29.6% yes 70.4% no	6.603	**0.010**	0.149
Family history of eating disorders	12.5% yes 87.5% no	13.2% yes 86.8% no	0.014	0.571	−0.007

BMI = Body mass index.

**Table 4 jcm-13-06952-t004:** Regression models for NSSI.

	B	SE	OR	95% CI	*p*
Model 1b: association between NSSI and sociodemographic variables					
Education	−0.033	0.068	0.968	0.847–1.105	0.626
Gender	−0.702	1.061	0.496	0.062–3.965	0.508
Occupation	0.297	0.385	1.346	0.633–2.861	0.440
Family history of psychiatric disorders	−0.944	0.360	0.389	0.192–0.787	0.009
Family history of eating disorders	0.251	0.531	1.286	0.454–3.641	0.636
Model 2b: association between NSSI and eating-related clinical features					
AN subtypes	1.091	0.439	2.978	1.260–7.039	**0.002**
Illness duration	−0.027	0.029	0.973	0.920–1.030	0.351
Age at illness onset	−0.150	0.060	0.861	0.765–0.969	0.818
BMI	0.324	0.134	1.383	1.064–1.798	**0.015**
Lowest lifetime weight	−0.010	0.048	0.990	0.901–1.087	0.829
Psychopharmacological treatment	1.007	0.535	2.737	0.960–7.805	0.060
Crossing over	−0.363	0.598	0.695	0.215–2.245	0.544
Previous hospitalizations	0.026	0.078	1.026	0.880–1.196	0.743
Model 3b: association between NSSI and comorbid clinical variables					
Suicide attempt	−0.895	0.494	0.409	0.155–1.077	0.070
Psychiatric comorbidities	−0.399	0.383	0.671	0.317–1.420	0.297
Personality disorders	−2.071	0.388	0.126	0.059–0.270	**<0.001**
Lifetime sexual abuse	−0.845	0.547	0.430	0.147–1.255	0.122
Model 4b: association between NSSI and eating-related psychopathology					
EDI-2 drive for thinness	−0.108	0.076	0.898	0.773–1.142	0.155
EDI-2 bulimia	0.079	0.057	1.082	0.968–1.210	0.166
EDI-2 body dissatisfaction	0.215	0.074	1.240	1.054–1.316	**0.004**
EDI-2 ineffectiveness	0.111	0.071	1.118	0.992–1.259	0.067
EDI-2 interpersonal distrust	−0.042	0.086	0.959	0.811–1.134	0.624
EDI-2 interoceptive awareness	−0.043	0.042	0.958	0.881–1.041	0.307
EDI-2 asceticism	−0.084	0.071	0.920	0.799–1.048	0.241
EDI-2 impulsivity	0.076	0.051	1.079	0.976–1.172	0.136
EDI-2 social insecurity	0.026	0.095	1.026	0.851–1.236	0.788
EDE-Q dietary restraint	1.138	0.534	3.122	1.095–8.900	**0.033**
EDE-Q food concerns	1.127	0.518	3.087	1.118–8.519	**0.030**
EDE-Q shape concerns	0.380	0.706	1.496	0.366–5.836	0.591
EDE-Q weight concerns	1.592	0.703	4.912	1.239–19.471	**0.024**
Model 5b: association between NSSI and general psychopathology					
BDI	0.052	0.040	1.053	0.974–1.138	0.193
STAI-state	0.027	0.030	1.027	0.969–1.090	0.365
STAI-trait	0.031	0.024	1.032	0.985–1.081	0.182

BMI = Body mass index; EDI-2 = Eating Disorder Inventory-2; EDE-Q = Eating Disorder Examination Questionnaire; BDI = Beck Depression Inventory; STAI = State-Trait Anxiety Inventory.

## Data Availability

The data that support the findings of the study are available from the corresponding author upon reasonable request.

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
