# Peer review of "Analysis of Suicidal Behavior in Adult Inpatients with Anorexia Nervosa: Prevalence of Suicide Attempts and Non-Suicidal Self-Injury and Associated Factors—Data Before and After the COVID-19 Pandemic"

_jcm, 2024, doi:10.3390/jcm13226952_

Round 1
Reviewer 1 Report
Comments and Suggestions for Authors
This paper focuses on anorexia. The abstract is well-structured!
Some points authors should take into consideration in their revision:
Please provide more information regarding the recruitment of the participants.
Authors can discuss the experience of these patients as described in qualitative studies (https://doi.org/10.1002/eat.20276 and https://doi.org/10.1080/13642537.2023.2278088and https://doi.org/10.1002/erv.774 and https://doi.org/10.1186/s40337-023-00736-9)
Why were these measurements used instead of others? Please justify?
Effect sizes are missing. Please add for all analyses.
A more critical discussion of the findings is necessary across the lifespan. For example, authors could also report on suicide and anorexia in older adults. Additionally, the work of therapists for these eating disorders must be discussed as in (https://www.ceeol.com/search/article-detail?id=936319) and expand more on the clinical and therapeutic future approaches. Please expand more and use relevant references to support.
Finally, the limitations must be presented in detail as apart form alcohol, it seems that numerous relevant variables were not included in the analyses.
Comments on the Quality of English LanguageMinor English language editing.
Author Response
We thank the reviewers and the editor for the thorough and insightful comments. We carefully considered each of your suggestions and applied the required revisions. We also revised and improved the English. We believe these changes significantly strengthen our work, thank you.
Rev 1
This paper focuses on anorexia. The abstract is well-structured!
Some points authors should take into consideration in their revision:
Please provide more information regarding the recruitment of the participants.
Thank you for the comment. We provided further details on the recruitment.
Authors can discuss the experience of these patients as described in qualitative studies (https://doi.org/10.1002/eat.20276 and https://doi.org/10.1080/13642537.2023.2278088and https://doi.org/10.1002/erv.774 and https://doi.org/10.1186/s40337-023-00736-9)
Thank you for this suggestion, we added a brief description of the qualitative studied at the beginning of introduction as follows: “Relatedly, beyond the diagnostic criteria, patients with AN qualitatively experience a wide range of negative emotions such as shame, and assign subjective meanings for anorectic behaviors that extend beyond the socio-cultural shared weight-centered paradigm. These meanings can encompass several conceptions of the illness including association with death (i.e., “wishing to starve oneself to the death; [5–7])”.
Why were these measurements used instead of others? Please justify?
The present study focused on eating-related psychopathology. EDE-Q and EDI-2 are the most frequently and internationally used self-report tools to investigate core and associated eating-related symptoms (e.g., restrain and perfectionism respectively). In our opinion, depressive and anxiety-related symptoms cannot be neglected so as they are often comorbid with eating disorders, again, we chose the most used and known measures for depression and anxiety. Moreover, all the questionnaires were validated in an Italian samples, thus they are statistically strong and valid. Finally, some studies investigating the associations between eating-related or general psychopathology and suicidality adopted these measures. We specified this last point for each measure.
Effect sizes are missing. Please add for all analyses.
Thank you for the suggestion, we added effect size to the analysis.
A more critical discussion of the findings is necessary across the lifespan. For example, authors could also report on suicide and anorexia in older adults. Additionally, the work of therapists for these eating disorders must be discussed as in (https://www.ceeol.com/search/article-detail?id=936319) and expand more on the clinical and therapeutic future approaches. Please expand more and use relevant references to support.
Thank you for this comment, we explored all the topics you suggested in the last paragraph of the discussion.
Finally, the limitations must be presented in detail as apart form alcohol, it seems that numerous relevant variables were not included in the analyses.
Thank you for the comment. We agree that some other relevant factors should have been addressed and we specified it in the limits as follows: “Third, some crucial aspects such as alcohol and substance abuse, previous psychotherapeutic sessions, and suicide ideation were not specifically investigated”. However, we think that the great number of factor analyzed is a strength of the study, since to our knowledge, no other studies considered such a number of variables. Moreover, we did not want to make the analyses too difficult adding a greater number of factors.

Reviewer 2 Report
Comments and Suggestions for Authors
A report of retrospectively analyzed data on 298 anorexia nervosa inpatients from 2014-2023 regarding their suicidality to consider any differences before and after the COVID-19 pandemic.
The study is well-conceived, well-analyzed, readable, and concise. The tables are clear and informative. The weaknesses regard citations to outdated references lacking the support of current research.
- The authors claim there is little research on their topic. Here is a Google Scholar search of the topic for research published since 2020: https://scholar.google.ca/scholar?as_ylo=2020&q=Analysis+of+suicidal+behavior+in+adult+inpatients+with+Anorexia+2+Nervosa:+prevalence+of+suicide+attempts+and+non-suicidal+self-+3+injury+and+associated+factors.+Data+before+and+after+the+COVID-+4+19+pandemic&hl=en&as_sdt=0,5. With “About 309 results”, the authors must compare and contrast their study with the most relevant returns of this search to demonstrate how their study adds to the literature.
- The Introduction should report the most recent research on a topic. Citations 1-5, 8, and 14-19 are outdated. For each of these citations that stand alone, please provide a supporting citation of research published since 2020.
- For each of the materials, please explain their selection in the text and cite current research demonstrating the continuing use of these materials in similar studies.
- The following citations in the Discussion are outdated and require supporting citations to current research: 8, 33, 36, 43, 44, 45,46, 47, and 50.
- Please end the report with suggestions for future research.
Author Response
We thank the reviewers and the editor for the thorough and insightful comments. We carefully considered each of your suggestions and applied the required revisions. We also revised and improved the English. We believe these changes significantly strengthen our work, thank you.
Rev 2
A report of retrospectively analyzed data on 298 anorexia nervosa inpatients from 2014-2023 regarding their suicidality to consider any differences before and after the COVID-19 pandemic.
The study is well-conceived, well-analyzed, readable, and concise. The tables are clear and informative. The weaknesses regard citations to outdated references lacking the support of current research.
- The authors claim there is little research on their topic. Here is a Google Scholar search of the topic for research published since 2020: https://scholar.google.ca/scholar?as_ylo=2020&q=Analysis+of+suicidal+behavior+in+adult+inpatients+with+Anorexia+2+Nervosa:+prevalence+of+suicide+attempts+and+non-suicidal+self-+3+injury+and+associated+factors.+Data+before+and+after+the+COVID-+4+19+pandemic&hl=en&as_sdt=0,5. With “About 309 results”, the authors must compare and contrast their study with the most relevant returns of this search to demonstrate how their study adds to the literature.
Thank for this precious suggestion, we carefully examined the literature and chose some relevant and recent studies to add to our work, both in introduction and discussion. In particular, we added some data from Arnold et al., 2022, Kirkpatrick et al., 2023, Warne et al., 2021, and Soullane et al., 2023. We think these reference added strength to the rationale of the study and depth to the discussion.
- The Introduction should report the most recent research on a topic. Citations 1-5, 8, and 14-19 are outdated. For each of these citations that stand alone, please provide a supporting citation of research published since 2020.
We updated several references through the text, thank you.
- For each of the materials, please explain their selection in the text and cite current research demonstrating the continuing use of these materials in similar studies.
All the questionnaires selected for the study are among the most used and validated tools to assess eating-related and general psychopathology. Moreover, the questionnaires are currently used to investigate the association between the specific symptoms object of the tool and suicidality; we specified it in the text, thank you.
- The following citations in the Discussion are outdated and require supporting citations to current research: 8, 33, 36, 43, 44, 45,46, 47, and 50.
Thank you for the comment. We updated the references accordingly.
- Please end the report with suggestions for future research.
Thank you for this helpful suggestion, we added future directions at the end of the manuscript as follows “On a theoretical standpoint, the present study may boost future researches on the topic: for instance studies with longitudinal design are needed and future works should deepen the relationship among the factors significantly associated to SA and NSSI. Moreover, studies on suicidality in AN in older adult are lacking; it is acknowledged that suicide in individuals older than 60 is a critical issue [74]. In this context it could be useful to deepen how suicidality develops in older patients with AN given the potential overlap of risk factors such as depression or additional cognitive impairment that could exacerbate pre-existing deficit in cognitive flexibility and decision making typical of AN and associated with suicide risk in older adults [74]. Finally, investigations on possible biological factors potentially involved in the association between eating disorders and suicidality are scarce.”

Reviewer 3 Report
Comments and Suggestions for Authors
Thank you for the opportunity to read this interesting paper. The work is valuable for clinicians and acts as a way into some further research. A challenge with this type of work is finding ways to parse out one aspect of a complex clinical picture - thus the results are correlational as opposed to causative. This is not surprising given how complex the clinical history and presentation can be with ED. This might be worth some further attention in the discussion.
Your notes on line 266/267 about AN and BPD faces an intersectional challenge in that the SA and NSSI rates for BPD is, in itself, high. Does AN make that worse? Your data does not appear to clarify this and thus I am wondering if it is wise to suggest AN enhances SA/NSSI in BPD or vice versa.
In the same area, you note the linkage between childhood sexual abuse (CSA) and ED (particularity AN) and SA/NSSI. Again, given that the linkage between CSA and SAS/NSSI is high, does AN make it higher?
I guess where I am going with this is suggesting caution about the strength of relationships between ED and SA/NSSI when co-morbid with disorders or traumas. I agree they do co-exist but is that the story (i.e. co-existence) as opposed to one making the other worse. I am concerned about the notion that they are necessarily compounding in the clinical picture when they may not be - the data presented does not allow that conclusion to be drawn.
This is good research - I am just cautious about the messages readers will take away.
I hope my comments make sense and are helpful. Thank you for your work.
Author Response
We thank the reviewers and the editor for the thorough and insightful comments. We carefully considered each of your suggestions and applied the required revisions. We also revised and improved the English. We believe these changes significantly strengthen our work, thank you.
Rev 3
Thank you for the opportunity to read this interesting paper. The work is valuable for clinicians and acts as a way into some further research. A challenge with this type of work is finding ways to parse out one aspect of a complex clinical picture - thus the results are correlational as opposed to causative. This is not surprising given how complex the clinical history and presentation can be with ED. This might be worth some further attention in the discussion.
We thank you for the comments. We followed your suggestion and briefly discussed the point you raised as follows: “In light of this, the present study represents an effort to isolate specific factors in the complex and entangled picture of eating disorders; future studies should keep on parsing out single variables involved in the clinical presentation of the disease trying to adopt a longitudinal design to seek for causative instead of correlational results”.
Your notes on line 266/267 about AN and BPD faces an intersectional challenge in that the SA and NSSI rates for BPD is, in itself, high. Does AN make that worse? Your data does not appear to clarify this and thus I am wondering if it is wise to suggest AN enhances SA/NSSI in BPD or vice versa.
This is an interesting issue. Unfortunately, our data are not able to clarify it since supposing that AN enhances suicidality in BPD or vice versa would be beyond data; future longitudinal studies could deepen this interesting point. We specified this in the text as follows: “Despite the cited studies seeming to suggest a role of borderline personality in enhancing suicidality in AN this causal correlation cannot be supported by our data; future longitudinal studies could clarify whether borderline personality disorder worsen suicidality in AN or vice versa.”
In the same area, you note the linkage between childhood sexual abuse (CSA) and ED (particularity AN) and SA/NSSI. Again, given that the linkage between CSA and SAS/NSSI is high, does AN make it higher?
This is a very interesting point too, but still we are not able to confirm the statements. We added to the discussion a warning to interpret our hypothesis cautiously since no empirical data support it.
I guess where I am going with this is suggesting caution about the strength of relationships between ED and SA/NSSI when co-morbid with disorders or traumas. I agree they do co-exist but is that the story (i.e. co-existence) as opposed to one making the other worse. I am concerned about the notion that they are necessarily compounding in the clinical picture when they may not be - the data presented does not allow that conclusion to be drawn.
This is good research - I am just cautious about the messages readers will take away.
You are right and your comments are really useful. We further stressed the point adding “Relatedly, both in the case of borderline personality disorders and childhood trauma, the picture is extremely complex and entangled, and the higher the clinical complexity, the harder the causal relationships to be drawn.” Thank you.
I hope my comments make sense and are helpful. Thank you for your work.

Round 2
Reviewer 1 Report
Comments and Suggestions for Authors
Most of the points raised have been answered, but not in an extended way in the main text, so authors need to discuss more each of the points. Furthermore, authors should discuss all suggested articles: Authors can discuss the experience of these patients as described in qualitative studies (https://doi.org/10.1002/eat.20276 and https://doi.org/10.1080/13642537.2023.2278088 and https://doi.org/10.1002/erv.774 and https://doi.org/10.1186/s40337-023-00736-9)
Author Response
We thank you the editor and the reviewer for the suggestions; we carefully addressed each point, thank you.
Most of the points raised have been answered, but not in an extended way in the main text, so authors need to discuss more each of the points. Furthermore, authors should discuss all suggested articles: Authors can discuss the experience of these patients as described in qualitative studies (https://doi.org/10.1002/eat.20276 and https://doi.org/10.1080/13642537.2023.2278088 and https://doi.org/10.1002/erv.774 and https://doi.org/10.1186/s40337-023-00736-9)
Thank you for your suggestions. We deepened the discussion on the points you raised. In particular:
- We expanded the discussion on qualitative studies we had already cited in the previous version of the manuscript both in the introduction and discussion. In the previous version of the manuscript we had not cited Traganzopoulou and Giannoulu (2023) because not strictly related to the topic of our work, but thank to your comment we added also this part to the introduction and to the discussion.
- We expanded the discussion on the lifespan as follows: “Moreover, studies on suicidality through the lifespan in AN are needed: studies indeed described not only static (e.g., gender) factors linked to suicide but also dynamic ones (e.g., substance use) that change in the different stages of the life [72,73]. For instance, the focus on older adults with AN could be insightful; it is acknowledged, indeed, that suicide in individuals older than 60 is a critical issue [74] and it could be useful to deepen how suicidality develops in older patients with AN given the potential overlap of risk factors such as depression or additional cognitive impairment that could exacerbate pre-existing deficits in cognitive flexibility and decision-making typical of AN and associated with suicide risk in older adults [74]”.
- We deepened the discussion on future research as follows: “From a theoretical standpoint, the present study may boost future research on the topic: for instance, studies with longitudinal design are needed and future works should deepen the relationship among the factors significantly associated with SA and NSSI. Moreover, studies on suicidality through the lifespan in AN are needed: studies indeed described not only static (e.g., gender) factors linked to suicide but also dynamic ones (e.g., substance use) that change in the different stages of the life [72,73]. For instance, the focus on older adults with AN could be insightful; it is acknowledged, indeed, that suicide in individuals older than 60 is a critical issue [74] and it could be useful to deepen how suicidality develops in older patients with AN given the potential overlap of risk factors such as depression or additional cognitive impairment that could exacerbate pre-existing deficits in cognitive flexibility and decision-making typical of AN and associated with suicide risk in older adults [74]. The exploration of patients’ feelings towards death and suicidality may also be explored with qualitative studies. Finally, investigations on possible biological factors potentially involved in the association between eating disorders and suicidality are scarce”.
